# A Biomorphic Model of Cortical Column for Content—Based Image Retrieval

**DOI:** 10.3390/e23111458

**Published:** 2021-11-03

**Authors:** Alexander Telnykh, Irina Nuidel, Olga Shemagina, Vladimir Yakhno

**Affiliations:** 1Autowave Processes Laboratory, Department of Radiophysical Methods in Medicine, Federal Research Center Institute of Applied Physics of the Russian Academy of Sciences (IAP RAS), 603950 Nizhny Novgorod, Russia; aleksander.telnykh@ipfran.ru (A.T.); olga.shemagina@ipfran.ru (O.S.); yakhno@ipfran.ru (V.Y.); 2Neurotechnologies Department of the Institute of Biology and Biomedicine, Lobachevsky State University of Nizhny Novgorod (UNN), 603022 Nizhny Novgorod, Russia

**Keywords:** biomorphic modeling, content-based image retrieval, recognition, cascade detection, adaptive boosting

## Abstract

How do living systems process information? The search for an answer to this question is ongoing. We have developed an intelligent video analytics system. The process of the formation of detectors for content-based image retrieval aimed at detecting objects of various types simulates the operation of the structural and functional modules for image processing in living systems. The process of detector construction is, in fact, a model of the formation (or activation) of connections in the cortical column (structural and functional unit of information processing in the human and animal brain). The process of content-based image retrieval, that is, the detection of various types of images in the developed system, reproduces the process of “triggering” a model biomorphic column, i.e., a detector in which connections are formed during the learning process. The recognition process is a reaction of the receptive field of the column to the activation by a given signal. Since the learning process of the detector can be visualized, it is possible to see how a column (a detector of specific stimuli) is formed: a face, a digit, a number, etc. The created artificial cognitive system is a biomorphic model of the recognition column of living systems.

## 1. Introduction

One of the topical directions in the study of living systems is the development of functional models that will give a better insight into the way information is processed by humans and animals. In this work, the technique and the process of constructing detectors for content-based image retrieval (CBIR) are correlated with the formation of active connections in the neuromorphic model cortical column. An intelligent artificial cognitive system, in which content-based image retrieval is performed in accordance with a given semantic dictionary (for example, character recognition), has been developed [1]. The developed recognition system is a variant of the model, where the interaction of structural and functional neural modules is similar to the architecture of columns in the human visual system. The potential and prospects of such modeling are revealed by visualization of the learning process on an example of different objects of the external world and by recognition of the given objects in different contextual situations.

The subject matter of the work is outlined above. There are different approaches to the development of content-based image retrieval systems, including bionic ones. The review is given in Section 2. On the way of developing such technical systems, researchers ask themselves how information is processed in living systems. An overview of long-term studies of the structural and functional organization of neurons in the cerebral cortex and the confirmation of the columnar organization of the visual system are given in Appendix A. Biomorphic models of neural networks, such as computational neuroscience models and neuroinspired machine learning models, are being developed. Some biomorphic neural network models, algorithms and systems are discussed in Section 3.

This paper provides an overview of quite a number of biomorphic information processing models. What for? We want to mark the place of the model developed by us in this succession. It is an adaptive model with a reconfigurable architecture and visualization of the learning process.

The system of content-based image retrieval as a biomorphic model of the formation and actuation of structural and functional modules of living systems during visual signal processing is described in Section 4. It is shown that the formation of detectors of objects of a given type during the learning process is a model of the process of constructing a column of the visual cortex, which is formed not only due to neurogenesis, but also due to the incoming afferentation. Object recognition in the presented images models the process of information processing in the neural columns of living systems—columns’ response to the presented stimulus. This is an illustration of the process of formation of connections of the model neural column of the recognition system as a result of presenting images from a training sample, which is an analog of an external signal of a real human visual system. This is how cortical columns are formed as structural and functional units of information processing in living systems as a result of afferentation by an external signal. The capabilities of the biomorphic model of the cortical column are considered in Section 5. The paper ends with the conclusions in Section 6, research prospects in Section 7, and a list of references.

## 2. Review of the Problem of Content-Based Image Retrieval

Currently, intelligent video analytics systems are being actively developed. Video analytics is the technology that uses computer vision methods to automatically obtain diverse information based on the analysis of a sequence of images received from video cameras in real time or from archive records [2,3].

Optical recognition of any images implies detecting and identifying an object, or classifying and determining any of its properties from the incoming image. Among the various types of classifications according to the level of significance and complexity, of special interest is content-based image retrieval [4,5], which gives an answer to the question: “what is shown in the picture?”

### 2.1. Scheme of Content-Based Image Retrieval

In living systems, the analysis of information coming through different sensory channels is organized hierarchically. Content-based analysis is also based on hierarchical image processing: analysis of primary features, analysis of secondary features, and content-based image retrieval, which is the pinnacle of the hierarchical image processing procedure. The problems of content-based image retrieval were reviewed in [6,7].

Primary features are represented in a high-dimensional space and have different levels of information content. From them, secondary features are formed with approximately the same level of significance. The main requirement for the analysis procedure is to select an optimal system of features and reduce the dimension of the feature space. In the learning mode, the classification system interacting with the operator, who marks the data, accumulates a bank of semantic concepts. At the top level of the pyramid, content-based image retrieval occurs. If the answer is ambiguous, the decisions made are ranked using the significance level score [8,9].

The methods for content-based image retrieval generally follow a typical scheme, but the methods of stage implementation are diverse. The problem of content-based image retrieval has a long lasting (over 20 years) history [8,9]. The first works date back to the 1980s. Since 2000, the following systems have been developed: content-based image retrieval with relevance feedback at the University of Illinois [10], PicToSeek [11], DrawSearch [4], and Viper (visual information processing for enhanced retrieval) at the Computer Science Department of the University of Geneva [12]. Such systems [13,14,15,16,17,18,19] are being improved by optimizing search procedures for images in specific large databases. In some systems, the feature space is very large.

The interactive dialogue with the user in CBIR systems is interpreted as a learning process with a teacher (supervisory training) simulating the mechanism of human perception of images [15,16,17].

One of the most challenging research problems in content-based image retrieval is bridging a significant semantic gap between the low-level functions of extracting elementary image features from pixel-by-pixel analysis (textures, colors, etc.) and semantic decision-making concepts. This includes texture query and image search based on visual perception. The research issues of image retrieval based on visual perception also include regions of interests detection (ROIs detection), image segmentation, relevance feedback, and personalized retrieval.

Supervised neural networks, methods of probabilistic classification and weighted Euclidean distances in a multidimensional space are used as means of forming weights of the irrelevant feature dimension in the feature space [15,17,18]. During the dialogue, feature weights are assigned to terminological concepts. As a result, a certain similarity function is constructed. However, the separating power of such a function is severely limited since, due to the complicated computations, not more than a quadratic form is usually used for constructing similarity functions.

Neural network models are well suited for this task, but require a large amount of training data for each new request. An alternative approach of unsupervised learning based on the use of self-organizing Kohonen maps was considered in [19,20]. The advantage of this method is that no interaction with a human operator is required in the process of training the system. On the other hand, the separating power of the recognition system reduces, due to the limited rules of map formation. In addition, for the operator to recognize the result of the retrieval, the automatically generated concepts must be interpreted, which is not always possible.

Today, deep learning technologies have significantly improved and simplified content-based segmentation algorithms [21,22,23], and are widely used in real life (analysis of satellite images, analysis of facial expression, analysis of medical images, etc.). Deep learning networks as biologically inspired systems are discussed further in Section 4.

The technology of content-based image retrieval is being actively developed. Bionic algorithms for semantic classification have been proposed along this path. Below is an example.

#### An Example of a Bionic Algorithm for
Content-Based Image Retrieval

The algorithm of content-based classification of JPEG images and the results of its modeling in the MATLAB software environment were discussed in [7,24,25]. The algorithm is based on three main ideas: (1) images are classified in a spectral feature space formed by a standard JPEG-format block coding procedure, which allows classification without image retrieval; (2) an economical hierarchical classification procedure is implemented: the content of the full image is a derivative of the content of image segments; and (3) the classification is performed according to reliable database precedents, thereby excluding any a priori assumptions about the configuration of the semantic class in the feature space.

An algorithm for adaptive image segmentation was proposed. The corresponding model makes use of the fact that, when perceiving an image, the human eye sequentially fixes its gaze on a number of characteristic features [26,27]. The point of eye-gaze fixation is the minimum area of the image that is sufficient for the human eye to extract its distinctive features. A JPEG image is analyzed assuming that the size of the gaze fixation point is determined by the size of one image block. The number of gaze fixation points depends on the level of complexity of the image, but in any case, this number is much smaller than the number of image blocks. The considered model has a limitation—only JPEG images can be classified because of the block organization of images.

According to the literature, some developers of content-based classification systems are looking for solutions in the field of biomorphic models for the software or hardware-software complexes to perform functional operations similar to their biological prototypes. In this case, the information processing systems themselves could be considered neuromorphic functional models of living systems.

## 3. Biomorphic Models of Neural Networks. Development of Algorithms and Technical Systems for Information Processing

### 3.1. From Experimental Data to Brain Theory

Brain research methods are being developed at a rapid pace. New experimental methods have appeared, for example, using light for registering brain activity, studying fluorescent proteins which encode light-sensitive ion channels, and others. Neurophotonics methods compete with electrophysiological techniques. Data on brain activity in various situations are constantly replenishing. The number of publications is growing: 298,934 articles in the PubMed database in the field of brain research appeared in 2019 (cf. 28,029 articles in 1969). However, according to Richard Axel, an eminent molecular biologist and neurologist, the 2004 Nobel Prize winner in biology and medicine, there is still no logic in turning a large number of facts of neural activity into theory (2019) [28].

The main tasks of brain research have already been distributed among the major programs. The creation of a theory of the brain is on the agenda. Let us consider two different approaches to this difficult task: the development of hypernet brain theory [28,29,30] and the development of formalized models for describing living systems [31].

It is necessary to obtain insight into the brain structure as a synthesis of interconnected and interacting functional systems distributed over a huge number of nerve cells in the brain. We believe that a possible approach of Russian researchers may be based on the theory of functional systems (TFS) developed in the 1930s to 1970s by the neurophysiologist P.K. Anokhin [32]. According to Anokhin, a functional system is a combination of the central and peripheral elements of the organism, which allow it to successfully perform one or another function.

#### 3.1.1. Mind as a Neural Hypernetwork

Many systems are innate, but many functional systems of a human develop, distinguishing one person from another. Each functional system is a group of cells controlled from the center (brain), which includes peripheral elements that make up the “I” of a person.

Such a network is comprised of neural networks, each of which is a functional cognitive group (COG) carrying evolutionary species knowledge or individual unique knowledge. The term COG was proposed by K.V. Anokhin. The mind in the framework of the brain theory is a neural hypernetwork, a highly ordered structure of the brain containing elements of our “I” [28,29,30].

The mind has a granular structure; it consists of cognitive elements—COGs. COGs mediate the information relationship between the integral cognitive agent and the environment.

COGs form stable cognitive links with each other—LIGs. LIGs reflect the cause-and-effect relationships of the elements and processes in the environment and in the relationships between the cognitive agent and the environment.

COGs and LIGs form a single cognitive network—the cognitome—which is the carrier of the subjective experience of a cognitive agent, that is, “I”. The mind is systemic and forms a unique structure of a personality.

Looking at brain research in terms of the fundamental task of constructing brain theory, one can understand how to interpret the latest experimental data [33,34].

#### 3.1.2. Development of Formalized Models for Describing Living Systems

Another approach to the development of the theory of the brain is the development of formalized models for describing living systems for common understanding and the development of a single language for describing the reactions of living systems to the perception of various sensory signals [31,35].

The mathematical models of subsystems, the mutual functioning of which allows analyzing a wide range of reactions inherent in living systems, are being developed. One of these subsystems describes possible states of a basic control recognition module that makes decisions based on the processed sensor signals. If the errors in its functioning exceed the specified threshold, a stress response trigger signal is generated. The other subsystem describes the mechanisms of formation and different levels of the development of three stages of the stress response, alternative variants of reducing stress and controlling sensory signal perception thresholds in the first subsystem. For this, additional variables corresponding to the experimentally recorded data and knowledge about the “fast” and “slow” stages are used in the model subsystems.

The performed research demonstrates the validity of the chosen model architecture and the possibility of using the results of its analysis as an adequate “language” of communication for researchers of living systems. The formalized models are important for understanding the meanings and consequences of unconscious perception through “image”, including sensation channels. They allow us to formalize the description of a number of processes, which were earlier interpreted ambiguously. Options for comparing dynamic modes of the integral model with known experimental data and interpreting them are actively discussed. The necessary condition for the construction of a neuromorphic model of an information processing system and its implementation in technical systems is that the developed software or hardware-software complexes should perform functional operations similar to their biological prototypes. A local neural network based on the vertical ordering of neurons in the cerebral cortex is the elementary structural and functional unit from which the matrix representation of the topology of combinations of many excited and unexcited areas is formed.

### 3.2. Homogeneous Distributed Neuron-Like System—A Model of a Cortical Column Firing during Feature Extraction

In 1972–1973, H. Wilson and J. Cowan [36,37] investigated a one-dimensional model of a network of inhibitory and excitatory neurons without regard to the change in the number of active connections between the neurons during excitation. S. Amari studied the dynamics of pattern formation in neural media with lateral inhibition [38]. These were the pioneering works on models of receptive fields of feature extraction—nonlinear convolutions with matrices of elements connected by a close nonlocal coupling function.

A model of a homogeneous distributed neuron-like system—a model of interconnected minicolumns—was presented in papers [35,39]. It is known that the neural column of visual zone V1 of humans and monkeys contains 260 pyramidal neurons, with about 1270 columns per mm^2^. All cells within one microcolumn are connected with the same receptive field. Adjacent microcolumns can be connected with different receptive fields. The axon emerging from the thalamus reaches 100–300 microcolumns. This means that the homogeneous approximation is valid. A homogeneous distributed system consisting of the same type of neuron-like elements with a close nonlocal coupling of the lateral inhibition type is a detector of objects of a given size. It performs the following operations: the convolution of image elements in a given neighborhood with a given coupling matrix (Figure 1 and Figure 2) and the comparison with a given threshold of the element. In such a system, which is a single-layer convolutional network, simple features of the original image are extracted, including contours (Figure 3), lines of preset directions, central axes of the image objects, objects of a given size (Figure 4), etc. Figure 4 shows the selection of objects of a given size by the example of selection of neuron somas and growing neural processes. The selection was carried out on a model of a homogeneous neuron-like system with different coupling functions, such as lateral inhibition.

### 3.3. Model of Associative Memory in Ensembles of Single Neurons

The fundamental issues of information coding and memory formation by single neurons and their ensembles in layered brain structures were formulated in [40]. Going deep into the details of the neural mechanisms of perception and memory, one expects that the complexity of the models will increase and the structural and functional relationships between brain structures and observed phenomena will become more sophisticated. Surprisingly, simple models of neuronal mechanisms of perception and memory show remarkable effectiveness.

The fundamental questions that can be answered using the model are the following:Selectivity;Clustering;The acquisition of memories.

Selectivity is the detection of one stimulus from a set. How is an arbitrary stimulus selected from a sufficiently large set so that one neuron from the neural ensemble detects this stimulus, that is, generates a response?

Clustering is finding a group of stimuli from a set. Within the set of stimuli, a smaller subset is selected, that is, a group of stimuli. What is the probability that the neuron detecting all stimuli from this subset keeps silent to all other stimuli in the set? It is expected that the solution to this problem will depend on the similarity of the stimuli within the group and their difference from the rest.

The acquisition of memories is the study of a new stimulus by comparing it with the already-known one.

The model of the external world should be supplemented with a model of a neuron and its adaptation. A possible solution to the three formulated fundamental problems within the framework of a simple classical modeling structure, where a neuron is represented by a perceptron with Hebbian-type learning, was shown in [41]. As expected in light of the stochastic separation theorems, the efficiency of a set of trained single neurons increases with the increasing amount of data.

The hypothesis of the existence of gnostic cells or grandmother cells [42], and hence, Jennifer Aniston cells, was confirmed in trustworthy neurophysiological experiments. “Grandmother of concept cells” can easily separate a specific multidimensional signal from all others without creating complex and essentially nonlinear rules. The theoretical papers in the field of artificial intelligence [41,43] confirm the hypothesis of the existence of gnostic cells [42] and explain the experiments on stimulation of small groups of cells capable of eliciting behavioral responses. Ensembles of weakly interacting, simple, neuron-like elements are an effective tool for solving essentially multidimensional and incomprehensible problems [41].

### 3.4. Structural Modeling of Neural Networks

#### 3.4.1. Topological Design of Fast Neural Networks

The authors of the works on structural modeling and topological design of fast neural networks [7,24,25] proposed a structural model of morphogenetic (self-similar) modular neural networks. Self-similarity, according to the authors, is a special case of morphogenesis. Algorithms of fast transformations are considered a variant of the model of a multilayer feedforward neural network. Self-similar networks allow for hierarchical expansion by additional planes with a dramatic increase in the number of recognized patterns. Self-similar networks with additional planes are architecturally similar to deep learning networks.

#### 3.4.2. Htm Structural Model of Interacting Columns

The organization of the cortical column [44,45,46,47] has inspired scientists to develop various architectures of technical systems, which are functional models of neuronal columns of information processing in living systems. For example, the self-taught associative memory (STAM) system is formed on the basis of a combination of online clustering and hierarchical predictive coding processes [48].

Hawkins et al. [48,49,50] constructed a system of continuous learning and recognition of any signals, using hierarchical temporal memory (HTM)—the machine learning technology aimed at repeating the structural and algorithmic properties of the cerebral cortex. HTM is a structural model of interacting microcolumns of the neocortex. For HTM training, one can feed continuous data flows (audio, video signals), and the system will learn, learn-after, and construct a data recognition model. Model initialization requires setting parameters, such as the number of columns, number of synapses in a column, threshold for synapse actuation, column overlap parameters, and others. Thus, the model of a column is specified in advance, whereas the process of its formation depending on external signal (afferentation) is obscure [49,50,51].

#### 3.4.3. Deep Learning Networks

Models inspired by studies of information processing in the brain are currently being actively developed. Here are just a few of them [22,52,53]. The problems of feature extraction from still images and video sequences are successfully solved by pretrained deep learning networks. Deep learning networks transmit the processed signal from the lower level to the higher level, and in this topo-topical organization of information transfer, there is a similarity between a technical recognition system and a living system.

For example, moving images can be recognized and object position on a video sequence can be predicted by means of a predictive neural network (“PredNet”) architecture that was inspired by the concept of “predictive coding” from the neuroscience literature. These networks learn to predict future frames in a video sequence, with each layer in the network making local predictions and only re-addressing the deviations from the predictions to the subsequent network layer [22].

The transformation of images on convolutional neural networks is similar to the passage of sensory signals through a system of receptive fields of columns into the higher zones of the cortex (coding). Convolutional neural networks are used to transform an input image to a smaller size through a series of convolutions. The output is then decoded through a series of transposed convolutions (e.g., fully convolutional network—FCN) [52]. Some convolutional networks have names that emphasize their similarity to biological prototypes, for example, Retina-Net [54]. The RetNet architecture is constructed as a multiscale convolutional pyramid. It has ascending paths, descending paths and lateral connections between the layers.

The design features of networks are outside of the scope of this overview. We only note that the procedure for their initialization is laborious, the architecture is extremely complex, and the process of eliciting connections and network growth is latent [55].

## 4. The System of Content-Based Image Retrieval—A Biomorphic Structural and Functional Model of Neural Columns of Information Processing in Living System

An intelligent artificial cognitive system was developed back in 2015 [1]. This system performs intellectual image analysis in accordance with a given semantic dictionary (for example, recognition of symbolic information) [56,57,58]. In the course of the development of an artificial cognitive system for content-based image analysis, a biomorphic model for the construction and interaction of structural and functional modules of living systems during visual signal processing was constructed.

The formation of detectors of objects of a given type in the learning process is a model of the process of building a column of the visual cortex, which occurs not only due to neurogenesis, but also due to incoming afferentation. Object recognition in the presented image models the process of information processing in the neural column of a living system, being the column’s response to the presented stimulus.

### 4.1. System of Content-Based Image Retrieval: General Description

The system of content-based image retrieval is organized as follows. Each concept of the dictionary corresponds to a cascade detector [59], sticking to strong classifiers based on nonlocal binary patterns [56]. The object properties are determined by an attribute detector also based on nonlocal binary patterns (Figure 5).

Objects in the images are detected by means of the multicascade detection technique. Each detector is a cascade of strong classifiers connected in series. The process of building detectors and their operation is a structural and functional model of information processing in a neural column, for example, in a cortical column of human and animal brain. The model enables tracing the process of formation (or triggering) of ascending connections in the column and obtaining a result of recognition, i.e., response to the presented stimulus in the form of an image. The layered structuring of a recognizing column is visualized.

### 4.2. System of Content-Based Image Retrieval: Algorithms

The specified objects in the image can be detected and recognized, using the approach proposed by Viola and Jones [59]. For developing efficient detectors, we make use of a modified census transform [56] for a rectangle of any size within the detection region.

How are the features of an original image extracted? The formation of code description is shown in Figure 6.

The images are in shades of grey. The average brightness of a rectangular fragment of the image is determined as follows:(1)$$\;\langle I_{F}\rangle =\frac{1}{w\cdot h}\sum_{x=x_{0}}^{x_{0}+w}\sum_{y=y_{0}}^{y_{0}+h}I\left(x,y\right),\;$$
where I(x,y) is the brightness of the pixel with x,y coordinates.

The area of formation of code description is divided into 9 equal parts, and the average brightness $\langle I_{f}\rangle$ of each part is analyzed in comparison with the average brightness of the entire area of formation of the code description. The average brightness $\langle I_{f^{n}}{\;}_{ij}\rangle$ of the area $f^{n}$ belonging to the fragment *F* is defined as follows:(2)$$\;\langle I_{f^{n}}{\;}_{ij}\rangle =\frac{9}{w\cdot h}\sum_{x=x_{i}}^{x_{i}+\frac{w}{3}}\sum_{y=y_{j}}^{y_{j}+\frac{h}{3}}I\left(x,y\right)\;$$
(3)$$\;c_{f^{n}}=\{\begin{array}{cc} 1, & \langle I_{f^{n}}\rangle \geq \langle I_{F}\rangle \\ 0, & \langle I_{f^{n}}\rangle <\langle I_{F}\rangle \end{array},$$
where 0≤i<3,0≤j<3,n=i·j. For each area $f^{n}$, the brightness is encoded according to the rule (Equation 3), where $c_{f^{n}}$ is the binary brightness code 0 or 1. In conformity with (Equation 3), for each image fragment, there is a set of nine code bits $c_{f^{0}}$ ... $c_{f^{8}}$. This sequence is considered to be code *c* of any fragment of image *I*. Thus, any rectangular fragment of the image is described by an integer number *C* having a width of 9 bits, i.e., 0≤C<512.

This means that in an array of 3 × 3 elements, there are $2^{9}=512$ nuclei. We use rectangular kernels of various sizes to encode graphic information. Examples of different kernels are shown in Figure 5. In neurophysiological terms, Figure 5 shows receptive fields of various types C-code of the receptive field) from the set of receptive fields $2^{9}=512$, which are used to encode information in the image. An example of overlapping of 3 × 3 rectangular arrays inside the aperture region is given in Figure 7 and Figure 8.

Thus, the considered features are defined as a kernel, 3 × 3 elements in size, reflecting the spatial structure of the image. Inside the kernel, binary coding of information {0;1} occurs, and the resulting binary templates may be boundaries, segments, saddle points, connection points, and so on.

#### 4.2.1. Weak Classifier

Each feature, that is, a nonlocal binary pattern, is a basis for the formation of a weak classifier, which is a binary function taking on the values of zero or one: zero in the absence of the sought object in the specified rectangular area of the image and one otherwise (Equation 3):(4)$$\;h_{k}=\{\begin{array}{cc} 1, & \hat{L}>0\\ 0, & \hat{L}=0 \end{array}$$
(5)$$\;\hat{L}=\frac{{\hat{\omega }}_{n}\left(u^{\prime }|\Omega =1\right)}{{\hat{\omega }}_{n}\left(u^{\prime }|\Omega =0\right)},$$
where $\hat{L}$ is the activation function of the weak classifier. To form the activation function, we use a decision rule based on the “maximum likelihood criterion” and its estimate (Equation 5), where ${\hat{\omega }}_{n}\left(u^{\prime }|\Omega =1\right)$ is the probability density estimate of the input signal value obtained from the analysis of the training sample, provided that the training sample represents samples of the “useful signal” and ${\hat{\omega }}_{n}\left(u^{\prime }|\Omega =0\right)$ is the estimate of the probability density of the value of the input signal obtained by analyzing the training sample, provided that the training sample is a background sample. Since the code has an integer nature, the estimate of its probability density distribution corresponds to the distribution histogram of the codes obtained from the analysis of the training sample.

#### 4.2.2. Strong Classifier

The so-called strong classifier (Figure 9) is formed using the AdaBoost learning algorithm [60,61], a set of features, and weak classifiers. A strong classifier is a binary function 0,1, including a decision-making threshold determined in the learning process. The learning process minimizes recognition error in the training database as follows:(6)$$\;H_{i}=\{\begin{array}{cc} 1, & \sum_{k=1}^{n}\omega_{k}h_{k}>\Theta \\ 0, & \sum_{k=1}^{n}\omega_{k}h_{k}\leq \Theta \; \end{array},$$
where $h_{k}$ is the weak classifier, $\omega_{k}$ is the weight of the weak classifier obtained in the learning process using the AdaBoost procedure, *n* is the number of weak classifiers, and Θ is the decision-making threshold.

As a result of training, a column (detector) is formed, which responds to objects of a given type. It is the key element of a search and recognition system. Objects in the image are detected, using the cascade detection technique. The detector is a cascade of strong classifiers connected in series (Figure 9). One of the tasks of improving the recognition system is the search for effective detectors, in which the IAP RAS team is involved [56]. The results of a computational experiment on the recognition of the numbers on the railway carriage or tank cars are presented [56]. To detect the number, a database of images of the trains passing near a video camera installed at the railway station was collected. The size of the training sample was 865 greyscale images for training and 274 greyscale images for testing. The false acceptance rate (FAR, type I errors) on the test sample was, on average, 10${}^{-8}$–10${}^{-6}$ for various detectors. The false rejection rate (FRR, type II errors) was 0.02 (for digits in the above experiment) to 0.25 (for other images).

## 5. Results

### 5.1. System for Content-Based Image Retrieval Is a Biomorphic Model of Cortical Column

Neocortex is known to be made up of six layers differing by the shapes and functions of the cells belonging to them, which are grouped into numerous columns (Figure A1) [62,63,64,65]. One of the functional operations of the cortical column is supposed to be the ability to respond to complex signals from the sensory system so that the signals corresponding to an object or a state activate the entire column, while the signals corresponding to another object or state do not activate the entire column. One or several cortical columns are required to recognize each object or state.

The proposed system of content-based image retrieval can be considered a multilayer neural network containing 7 layers. These include an input layer (integral image), a polymorphic layer (encoding brightness, gradients, texture and color), a layer of weak classifiers, a layer of strong classifiers, a hidden layer (provides spatial connection of solutions of cascades of strong classifiers), and an output layer (contains the locations of the found objects, their properties and the results of pixel-by-pixel segmentation).

The image arrives at the input layer.Polymorphic layer (6)—the polymorphic layer consists of neuron-like elements (neurons), which summarize information from a certain area of a rectangular image of arbitrary size. A neuron-like element is a formal threshold neuron that sums up incoming signals and has a nonlinear threshold activation function [35,39,66,67,68]. Each neuron-like element in this layer can be connected to a different number of neurons within the layer, from 4 to 1024. The function in the system is coding brightness, gradients, texture and color.The inner layer of pyramidal cells (inner pyramidal layer 5) contains neurons connected to a different number of neurons of the polymorphic layer (from two to nine), forming its receptive field, and, depending on their state, generates a code description of this state. The function in the system is the layer of weak classifiers.The code description of the state of the neurons of the inner pyramidal layer (5) is transmitted to one or more neurons of the inner granular layer (4), which, depending on the state of this code, are activated exclusively for a given type of signal. The “meaningful” activation of the neurons of this layer becomes possible as a result of training the entire neural network. The function in the system is the layer of strong classifiers.The neurons of the outer layer of pyramidal cells (outer pyramidal layer 3) are the classical artificial neurons of the “integrate-and-fire” type which are connected with several neurons of the previous layer. The function in the system is that the hidden layer provides spatial connection of the decisions of cascades of strong classifiers.Neuron-like elements of the outer granular layer (2) are also classical artificial neurons of the “integrate-and-fire” type, but they collect information from neurons of the previous layer and from neurons of their own layer, which belong to other cortical columns. One neuron is present in this layer as a rule. However, the case of two neurons mutually reinforcing or suppressing the activity of the entire hierarchy is also considered. The function in the system is the output layer, which contains locations of the found objects, their properties and the results of pixel-by-pixel segmentation.Elements of the sixth layer (1) are axon trees of neurons of the fifth layer (2), which transmit activating or suppressing signals to adjacent columns.

Table A2 shows that the characteristics of the developed cortical column model correspond to the known neurophysiological data on the structure and properties of its biological prototype. It is a visual biomorphic model of the formation and functioning of columns (information processing units) in living systems. On the one hand, such a model illustrates the processes of information transmission and processing in the human and animal brain. On the other hand, it has all the characteristics of a technical system for semantic image analysis.

### 5.2. Visualization of the Growth of the Cortical Column in the Learning Process and Column Firing in the Recognition Process

The functioning of a cortical column is investigated on an artificial column, which allows visualizing the layered structure of the neural network in the course of learning and the result of column activation during recognition. Visualization of the work of an artificial cortical column is a dynamic process in the course of which a given fragment of the image is analyzed. The visualization system displays the active states of the elements of the artificial cortical column at the current moment of time simultaneously with the result of its work in the original image. The column activity is visualized in 2D and 3D representation. In the process of learning on the training set, one can observe the growth of the detector column. The criterion for completing the formation of a column is the achievement of the required recognition accuracy.

In addition to visualizing the internal content of the column, it is also possible to retrieve information about the structure parameters, such as the following:Objective;The number of neuron-like elements in different layers;Thresholds of firing neuron-like elements in different layers;Activation functions of neuron-like elements;Neuron-like elements distribution over the field of view.

Different visualization modes of column operation are shown in Figure 10, Figure 11 and Figure 12.

### 5.3. Capabilities of the Biomorphic Model of the Cortical Column

A typical artificial biomorphic model of a cortical macrocolumn contains about 1000 neurons of various types and performs a strictly defined functional operation: the detection of objects of a given type from a definite angle of view.

A simplified model of a neural network inside a cortical column and of a section of the cortex consisting of such artificial columns designed to process incoming sensory information in the form of an image is constructed. As a result of the operation of such a structure, a content-based description of the processed image is formed, which is represented by the activity of various elements of the modeled section of the cortex. The activity of various columns of such an artificial structure corresponds to the recognition of one or another object in the image, and the determination of its properties. Each individual column of the model corresponds to one object from the semantic dictionary known to the recognition system. A semantic dictionary is a collection of objects viewed from a specific angle. The artificial structure is organized as a network of vertically oriented columns with clear functional roles of the layers within them.

The extraction of semantic features, excitation or inhibition of other columns involved in the processing of incoming visual information, formation of new connections between neurons, changes in the weights of existing connections, and extraction of the most essential features of the recognized objects are provided within each column.

An important feature of the presented model is that the structure of the cortical column has a universal nature and allows, using a small set of neuron types, analyzing images aimed to detect objects of various types and identify their characteristic properties. The types of neurons differ by the type of receptive fields. Examples of different receptive fields are presented in Figure 5.

Examples of objects of various types to which the system responds include faces of people, pedestrians, cars, license plates, individual numbers of figures, etc. Object properties are color, size, location, gender, age for humans, etc.

In the developed system of semantic image analysis, we obtain detectors—columns that are fired only in the presence of signals of a certain type. This system is a biomorphic model of interacting hierarchical columns of the same type. It includes several types of neuron-like elements working at their hierarchy level performing simple operations of transforming information.

It is shown that such a model of identical elements grouped layer by layer is able to recognize objects of given types and their properties (selectivity and content-based classification of images). The biomorphic artificial model of the cortical column is designed to detect in the image objects of specified types. Each object from the real world corresponds to one or more artificial cortical columns, which are activated if a signal corresponding to the object of a given type is fed to their input. Areas of activity of cortical columns trained for various objects of the real world are shown in Figure 13.

The present-day scanning system feeds fragments of an image to the input of a set of artificial cortical columns. The fragments eliciting artificial neurons in all layers of the cortical column receive a signal that they contain an object associated with the activated column. In this way, analysis of the activity of a set of artificial cortical columns can provide semantic description of the image consisting of the set of activity of the artificial cortical columns connected with the location of different objects in the image and their size.

The developed software package providing visualization of the operation of an artificial biomorphic macrocolumn enables studies of its learning. For example, the influence of the properties of neuron-like elements (receptive fields and activation functions) on the accuracy of recognizing objects of a given type, suppression of errors of the first and second kind in the process of formation and functioning of a macrocolumn may be investigated. Note that errors can be suppressed without retraining the entire system, but exclusively due to the growth of the microcolumn “in depth and in width”, that is, as a result of involving new elements in the structure of the column.

In addition to retraining a neural network during operation, the software package makes it possible to study the speed and efficiency of the entire system by estimating the number of neuron-like elements involved in the operation of the entire system during recognition. In other words, the activity of the column during observation of false objects is assessed, which allows drawing conclusions about the ways to optimize either the column model itself or the training procedure for minimizing this activity. The developed software package enables the user to change and control the system parameters (see Table 1).

Available value for **FileName** parameter

**Face.xml**—a cortical column that responds to a person’s face from the front.**FaceHalfProfile.xml**—a cortical column responding to a human half-face.**CarSide.xml**—a cortical column responding to a car from the side.**CarBack.xml**—a cortical column responding to a car from behind.**CarFront.xml**—a cortical column responding to a vehicle from the front.**CarHalfProfileBack.xml**—a cortical column responding to a vehicle from the rear side.**CarHalfProfileFront.xml**—a cortical column responding to a vehicle from the front side.**PedistrainFront.xml**—a cortical column responding to a pedestrian from the front.**PedistrainBack.xml**—a cortical column responding to a pedestrian from behind.**PedistrainProfile.xml**—a cortical column responding to a pedestrian from the side

## 6. Conclusions

Compared to other models of deep learning neural networks with a laborious initialization procedure, the developed system itself in the learning process forms the architecture of the column, which is an object detector. The column continues to grow until the specified accuracy is achieved. The process of forming the detector is visualized.

The developed artificial cognitive system is presented as a biomorphic model for the construction and interaction of structural and functional modules of living systems during processing visual signals. It is shown that the formation of detectors of objects of a given type in the learning process is a model of the process of building a visual cortex column formation in living systems not only due to neurogenesis, but also due to incoming afferentation. Object recognition in the presented images models the process of information processing in neural columns of living systems—the column’s response to the presented stimulus. Since the learning process of the detector is visualized, one can trace the formation of a column detector of specific stimuli: face, figure, number, etc. Thus, in the course of the development of a hierarchical system for semantic image analysis based on multistage detection, there occurs a metamorphosis: it appears that the technical system [1,56,57,58] can be considered a biomorphic model of structural and functional modules for information processing in living systems.

## 7. Research Prospects

The developed software package is actively used for recognizing objects of a given type in static images and video fragments. The following directions of further research are possible: the development of a technical recognition system and its practical application for recognizing objects of a given type; the development of new detectors of objects in the image (smoke detector, emergency situations, for example, fights, etc.); the illustration of a model of the process of formation of “conceptual cells”; and the search by the literature data for new neurophysiological and morphological data on column organization and bringing the results of the model column functioning in accordance with them (Table A2).

It is planned to simultaneously connect several interacting detector columns to increase the recognition accuracy and to organize a more complex scene analysis than the search for objects of a given type.

It is planned to compare the recognition accuracy of the developed system with the compound-scaled object detection model YOLOv5 [69], using standard image databases.

In light of the promising goal of constructing brain theory, visualization of the work of artificial neural networks is a highly topical task. For example, the “Transparent Brain” project is planned at the Institute for Advanced Brain Research of the Moscow State University [28]. For solution of the visualization problem, work with artificial neural networks should be organized as it is done in neurobiology: make the brain transparent to understand what happens inside the entire system and understand its external behavior. What will happen if we have the opportunity to control individual elements from the inside to see what such a system produces at the output?

Different tools for visualizing and analyzing neural networks used in the development and validation are currently being developed in machine learning. An intensively developing product is OpenVINO [70], which allows efficient formation of neural networks on various Intel architectures. One of the topical tasks is the development of software for visualizing the results of checking the accuracy of neural networks. With a little more time, deep learning networks will cease to be a black box. Our work on the formation of a biomorphic column for image detection and content-based analysis is in the mainstream of this research.

## Figures and Tables

**Figure 1 entropy-23-01458-f001:**
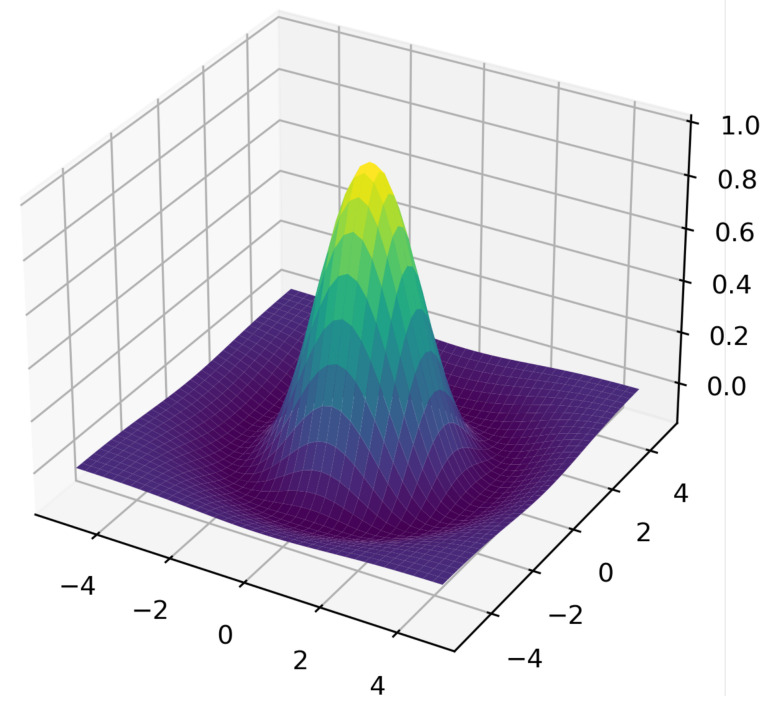
Coupling function of the lateral inhibition type.

**Figure 2 entropy-23-01458-f002:**
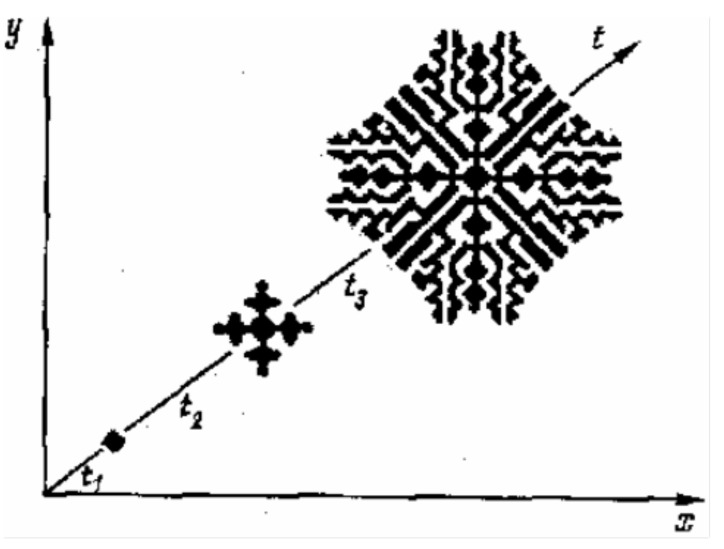
Examples of pattern formation in a homogeneous neuron-like system.

**Figure 3 entropy-23-01458-f003:**
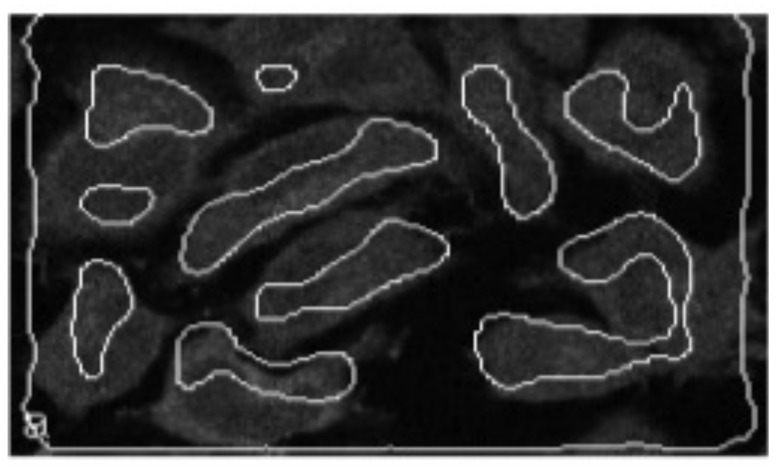
Extraction of contours from the initial cell image.

**Figure 4 entropy-23-01458-f004:**
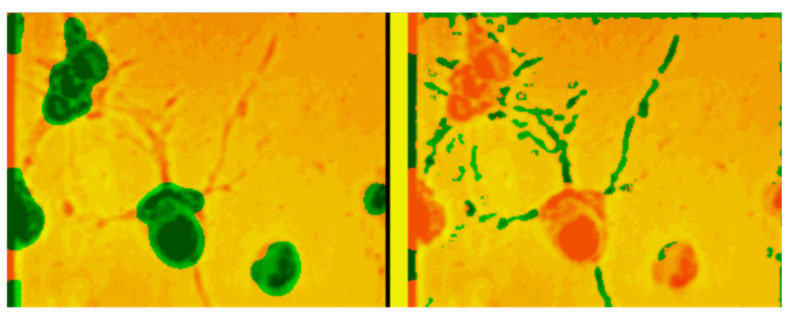
Extraction of neuron somas and neural processes as objects of a given size from the initial image.

**Figure 5 entropy-23-01458-f005:**
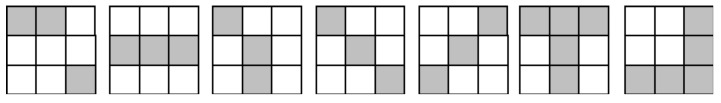
Examples of binary patterns (receptive fields in neurophysiological terms) used to encode information in an image. C-codes of receptive fields for pictures from left to right are equal, respectively, to 385, 56, 274, 273, 84, 466, 79.

**Figure 6 entropy-23-01458-f006:**
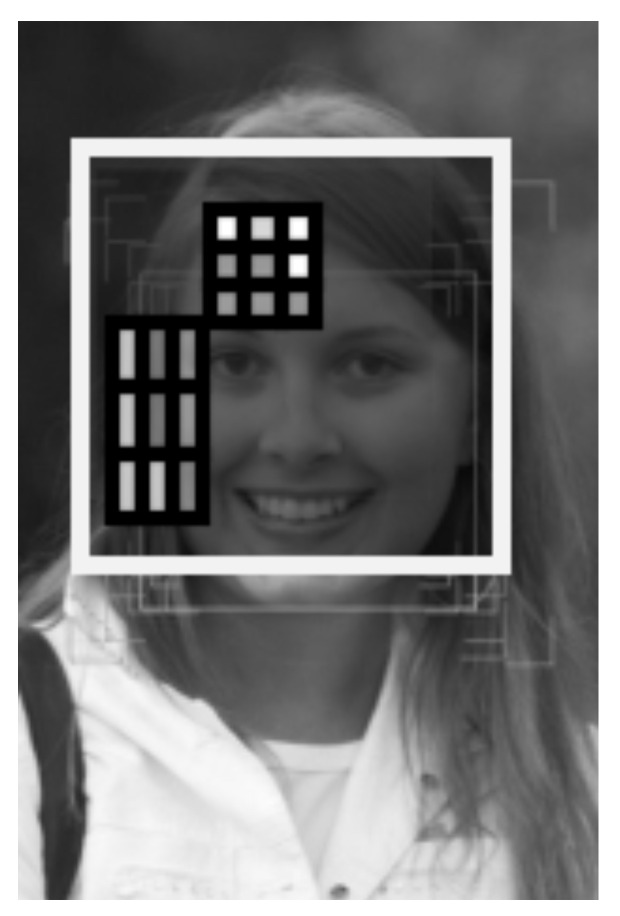
Formation of code description of a rectangular image area. The area of formation of code description is divided into 9 equal parts, and the average brightness $\langle I_{f}\rangle$ of each part is analyzed in comparison with the average brightness of the entire area of formation of code description.

**Figure 7 entropy-23-01458-f007:**
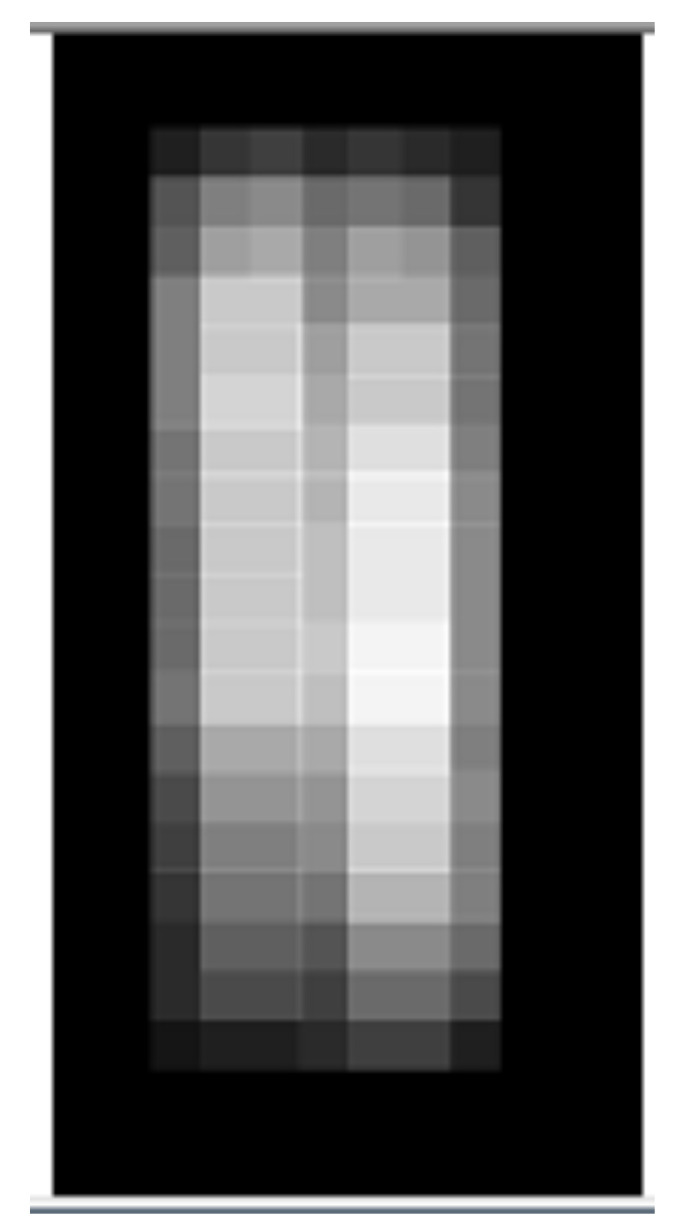
Example of overlapping of 3 × 3 arrays within the aperture.

**Figure 8 entropy-23-01458-f008:**
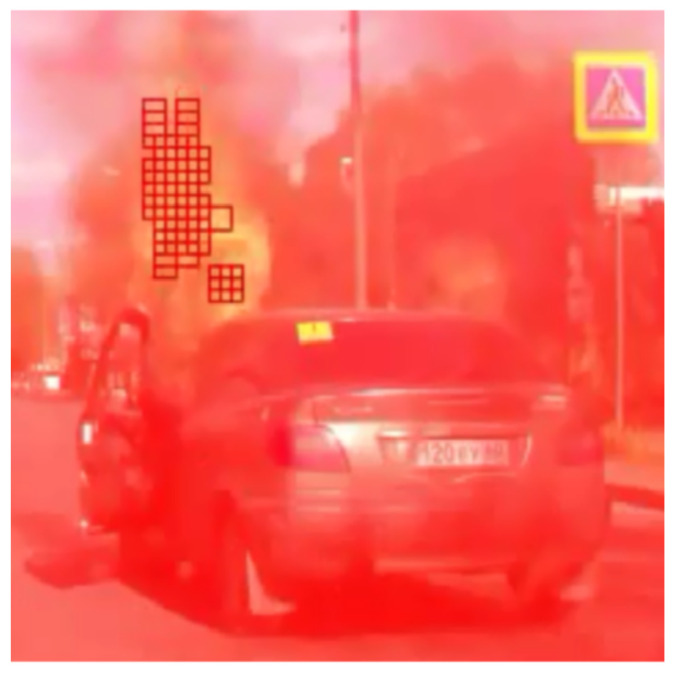
Example of overlapping of 3 × 3 arrays within the aperture in the real image.

**Figure 9 entropy-23-01458-f009:**
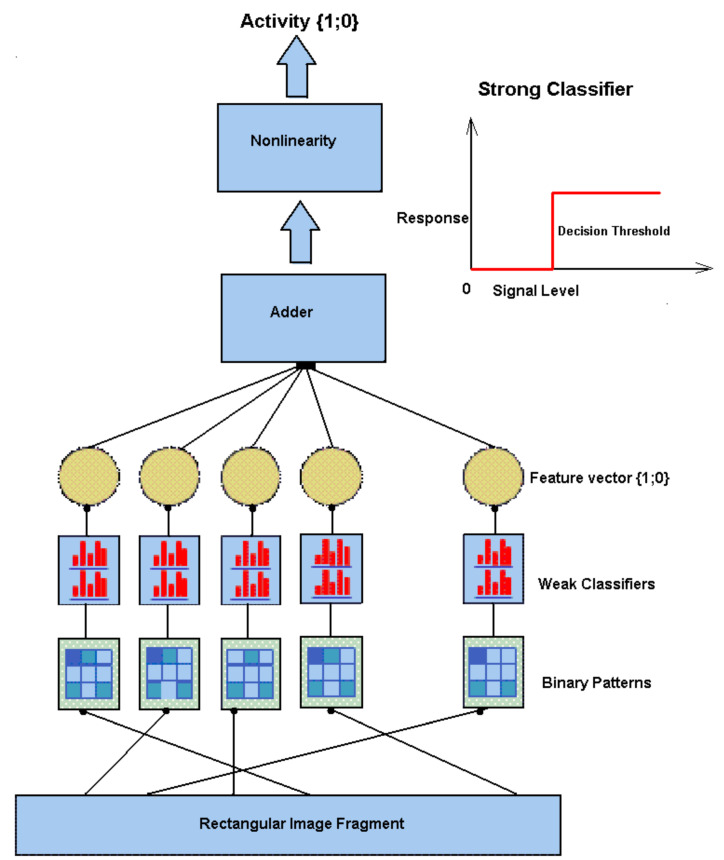
Strong classifier scheme.

**Figure 10 entropy-23-01458-f010:**
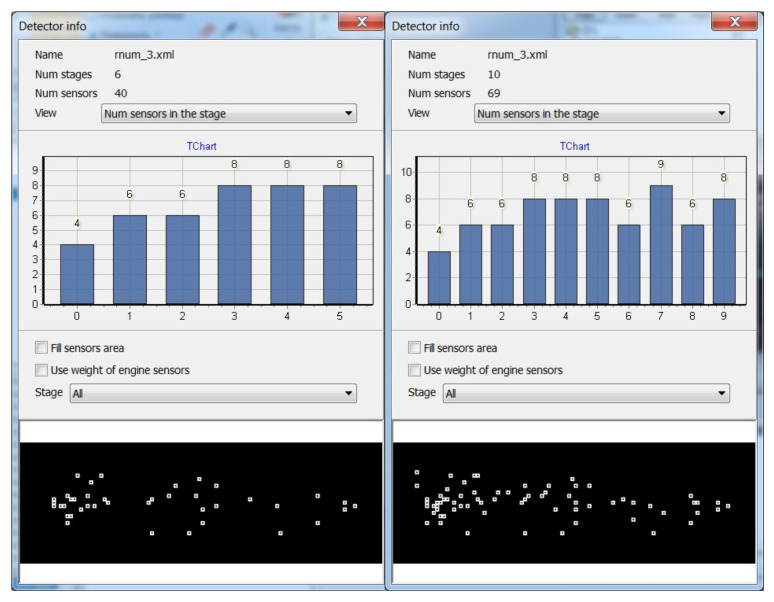
Top of figure: the number of elements in the outer pyramidal layer obtained in the learning process. Bottom of figure: example of displaying the distribution of the inner pyramidal layer elements obtained in the learning process for a given object.

**Figure 11 entropy-23-01458-f011:**
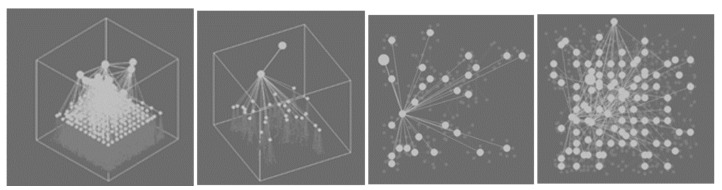
Example of displaying mapping the entire column in 3D representation; example of displaying a given element and its connections in 3D representation; example of displaying a given element and its connections in 2D representation; example of displaying several elements and their connections in 2D representation.

**Figure 12 entropy-23-01458-f012:**
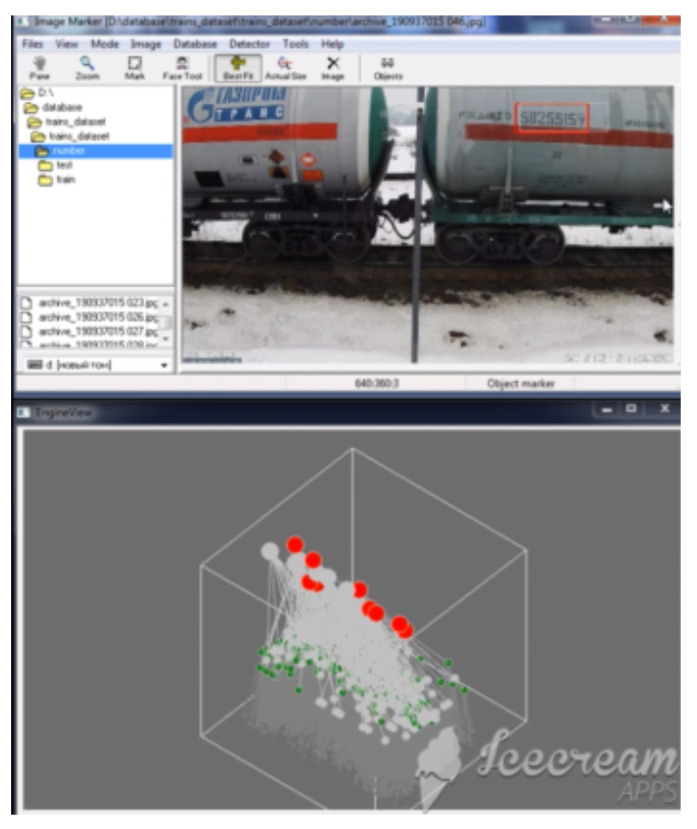
Visualization of the work of artificial cortical column in the course of object recognition. Active elements (lighted points) of the inner layer are shown in green among many inactive (grey) elements; the active elements of the pyramid are shown by red circles in the top layer.

**Figure 13 entropy-23-01458-f013:**
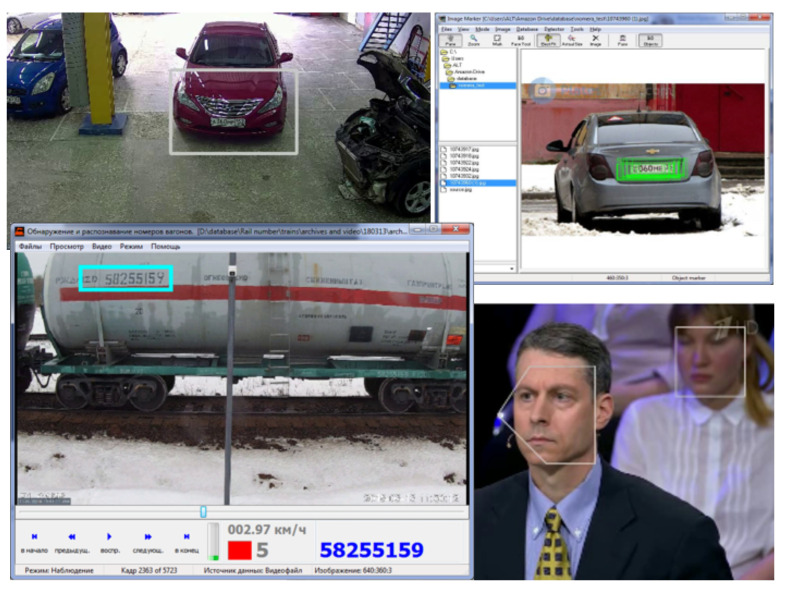
The response of cortical columns to different objects of the real world: cars, vehicle registration numbers, railway carriage numbers, and human full faces and profiles.

**Table 1 entropy-23-01458-t001:** The system parameters.

№	Parameter Name and Purpose	Available Values
1	**FileName** **File name** Each model of an artificial cortical column is in a file; to start working with it, the system must load it into memory.	**Face.xml** **FaceHalfProfile.xml** **CarSide.xml** **CarBack.xml** **CarFront.xml** **CarHalfProfileBack.xml** **CarHalfProfileFront.xml** **PedistrainFront.xml** **PedistrainBack.xml** **PedistrainProfile.xml**
2	**ImageName** **Image file name** Each artificial cortical column model works with an image file.	Any **jpeg** or **png** image. If the image is not specified, then the work of the cortical column cannot be visualized in dynamics. Only its internal representation is available.
3	**Rect** **The rectangle** in which the search for an object must be made	Any rectangular fragment of the image ImageName. If it is not specified, then the search is performed over the entire image.
4	**ObjectType** **Object type** Each of the artificial cortical columns can only respond to one type of object. This parameter indicates what type of object is associated with this column. The parameter is read-only.	Face, vehicle, pedestrian
5	**NumStages** The number of neurons in the pyramidal layer. The parameter is read-only.	0, 1,…, 100
6	**NumSensor(K)** The number of neurons of the inner granular layer associated with the K-th neuron of the pyramidal layer	1, 2,…, 1000
7	**Layer(N)** Display the N-th layer in the column.	True/false. N=0,… ,3
8	**Stage(N)** Display the N-th neuron in the pyramidal layer with all connections.	True/false. N=0,… ,NumStages
9	**3DView** Display the column in 3D.	True/false.

## Data Availability

The data used in this study are openly available in [1].

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
