# Peer review of "A Biomorphic Model of Cortical Column for Content—Based Image Retrieval"

_entropy, 2021, doi:10.3390/e23111458_

Round 1

Reviewer 1 Report

The paper proposes a neuro-inspired method for content-based image retrieval. The neuro-inspiration comes from the notion of cortical columns in the mammalian brain. 

Overall, I found this paper very interesting and I believe that it can be a valuable contribution when published. 

The problem however is that, in its current structure, the paper is rather confusing and without a clear focus. Let me elaborate:

The paper includes section-3, which is a long (several pages) overview of various theories/experiments about cortical columns and other brain-related knowledge. That section is quite broad -- for example in Table-1 it provides a summary of all the large brain-research programs around the world (the reader is left wondering: how is this related to the focus of this paper)? There are several other subsections or paragraphs in section-3 that are only marginally relevant to the paper's contribution, which is the content-based image retrieval model. 

Then, section-4 is also equally broad and "out of focus" providing an overview of either computational neuroscience models (such as Wlson/Cowan) or neuroinspired ML models (such as HTM). Again, the reader is left with the question: is this a survey paper or a specific technical contribution? And if it is the latter, how is this broad overview of section-4 related to the contribution?

The proposed model is described in section-5, which is actually quite short relative to the overall paper (a couple of pages). The model itself is pretty straightforward and the reader (at least this reviewer) is left wondering: if we ignore any hypothetical connections with neuroscience, what is the merit of the proposed model relative to the much more powerful approaches used by deep learning models today? 

Section-6 is called "Results" but it is very limited in terms of experimental results and it does not present comparisons with any state-of-the-art models from deep learning. Plus, a big part of that section (6.1) is not really about Results -- but about the connections between the proposed model and neuroscience. 

Here is my recommendation to the authors:

It would be good to split this paper into two different contributions/papers:

1) a paper that reviews what is currently known about cortical columns written for a computer science/ML audience. This survey paper would be useful to authors of neuro-inspired ML models. Instead of being only a survey, the authors can also discuss in this paper certain properties of cortical columns that may be relevant to the problem of image classification or content-based image retrieval. 

2) an ML-focused paper that proposes and evaluates the content-based image retrieval scheme. That paper may have some brief section about the underlying connections with neuroscience -- but that will hopefully be short and very focused. On the other hand, this second paper should be deeper in terms of how the proposed model compares with state-of-the-art ML models that solve the same problem. I recommend that the authors submit this paper to an ML conference or journal so that it is evaluated by the right audience. 

I would be interested to review either of these two papers if the authors agree to restructure their work in the proposed manner. 

Author Response

Dear Reviewer, thank you very much for the very helpful comments. Thanks to them, the quality of the material has improved significantly.The manuscript has been restructured in line with the Reviewer’s comments. The response to all the points of the review and the detailed description of the modifications are attached. Kind regards,
Dr. I. Nuidel

Reviewer 2 Report

Authors presented very interesting paper for vry important problem of brain work explanation.

There are several comments on English (typos).

Line 44. “presentating” should be “presenting”

Line 205. “Invesigation” should be “Investigation”.

Line 243. "vertical and angled lines overlap" should be "vertical and angled lines are overlaped".

Line 360. Full stop.

Line 413. “onnected” should be “connected”.

Line 552. “each area 0,1” should be “Line 552. “each area 0 or 1”. The same in other places.

Line 570. It is not necessary to add comma before "and" because you listed two items only.

Comment to cntent.

1. Line 93. Meaning of word “weights” is not clear. Do you mean weights of NN? DO you mean weights of weighted Euclidean distance? I think that the first but it should be clear without guessing.

2. Line 111. Authors state that problem is not solved yet. It is absolutely true. Authors do not discussed question about existence of unique absolutely correct solution of this problem. I am not sure that such solution exist.

3. Line 136. Usage of asterisk in image size is not good idea. There are two standard spelling: 100-by-150 or 100x150 (x should be \times).

4. Lines 131-133. Description of algorithm is inacceptable. It can be used less level of details or essentially much details. Now description produces many questions. “entire image consists of 21 points” means that you consider images with size 3x7 or 7x3. “segment polygon consists of 14 points” Why segmenting polygon contains 2/3 points of entire figure? Statement “polygon of the segment’s continuous area consists of 7 points” means that it is row or column only. Why it cannot be rectangle?

5. Line 158. Authors of papers 29 and 30 are different. It is necessary to use plural form: “Spanish neuroscientists”

6. Line 200. “(not more than 10 m/s)”. Maybe “(not less than 10 m/s)”? Otherwise term “slow” become strange.

7. Figure 1. It is stated that "Triangles are pyramidal cells." But it is not described what are rounds. Are rounds anothe type of cells? Statement that "interneurons of the corresponding levels of the cortex and inhibitory connections are shown in gray" is wrong because of at least part of interneurons connections are shown in black.

8. Figure 2. This figure required more explanations. It looks like that totally this column contains 4 pyramidal cells and 4  round cell. Some of connections is presented several times. For example, K25, K142, K125, K166. This figure required usage of much time to dechipher. You can use this representation but at least explain what do you mean in headers of four column figures? Is it different time?

9. Line 248. Please use sigh \times instead of *, because the last means complex conjugate.

10. Line 277 includes duplicate of line 278.

11. Table 1. Row 3. MINDS is abbreviation of the following part. It should be indicated. For example by adding brackets: (Brain Mapping by Integrated Neurotechnologies for Disease Studies).

12. Line 339-340. “Let’s consider two different approaches to this difficult task: [64,69,70] and [71].”. It looks like bad joke. It is necessary at least add some names of these approaches. “Let’s consider two different approaches to this difficult task: Name_of_approach_1 [64,69,70] and Name_of_approach_2 [71].”.

13. Line 361. “COGs form stable cognitive links with each other”. Really with “each other” or simply with “some other”?

14. Figure 3. Maybe grey scaled colouring of surface become this figure more readable, but I am not sure. From the other side, entropy is online journal and colour figures are acceptable. Now it is very difficult to read this figure.

15. Lines 421-422 and figures 5 and 6. Are contours and images of a given size extracted from the same original cell image? Maybe presenting of original cell image and then contours and objects of given size extracted from the same image will be more illustrative.

16. Formula 1 and 2. It is necessary to remove “x=” and “y=1” from superscript of summation (top label).

17. Function “brightness” is vell defined for grey scale images only. There are some problems with definition of brightness for colour images. It is preferable to state explicitly that authors consider grey scale images.

18. Formula 2 Left side MUST be indexed by i and j: \left<I_f^n\right>_{ij}.

19. Formula 3. Please check indices.

Author Response

Dear Reviewer, thank you very much for the very helpful comments.

Thanks to them, the quality of the material has improved significantly.

Kind regards,
Dr. I. Nuidel

In accordance with the remarks of the Reviewer, the structure of the article has been revised. All mentioned comments on English (typos) and comment on the content have been taken into consideration. The authors appreciate the Reviewer’s remarks that contributed to improving the manuscript.
